# Assessment of palladium concentration in drill cores using laser-induced breakdown spectroscopy (LIBS)

**François Vidal** [1]*, **Samira Selmani**[1], **Ismail Elhamdaoui**[1], **Nessrine Mohamed**[2], **Paul Bouchard**[3], **Marc Constantin**[2], **Mohamad Sabsabi**[3]

**1** Institut National de la Recherche Scientifique, Centre Énergie Matériaux Télécommunications, Varennes, QC, Canada, **2** Département de Géologie et de Génie Géologique, Université Laval, Québec, QC, Canada, **3** National Research Council of Canada, Boucherville, QC, Canada

* francois.vidal@inrs.ca

## Abstract

Laser-induced breakdown spectroscopy (LIBS) appears to be a promising technique for rapid on-site assessment of precious metal concentrations in ores. However, a number of issues need to be considered for the optimal use of this technique in practical situations. This article focuses on the number of measurements (i.e., spectra or laser shots) required to assess the mean palladium concentration in drill cores from the Lac des Iles mine (Ontario, Canada). We have performed a probabilistic study of the accuracy of the mean palladium concentration obtained by LIBS as a function of the number of measurements at random locations. For this purpose, we first used the results of a detailed laser scan of the core surface and then a mathematical model of the probability density of the palladium distribution to explore the parameter space, in particular the effect of noise on the measurements. We show that for the 1-meter-long, 2-centimeter radius quarter core samples analyzed, a few thousand randomly sampled locations generally provide an assessment of the palladium concentration within useful confidence limits. For a typical laser repetition rate of 50 Hz, such an analysis is a matter of minutes compared to hours or days using conventional methods.

## 1 Introduction

Precious metals, including palladium, are commercially mined in concentrations of a few parts per million (ppm) or less. Mine samples are currently analyzed using conventional wet chemistry, fire assay, ICP [1] or atomic absorption [2] techniques after a laborious, time-consuming and energy-intensive sequence of grinding, homogenization and dissolution that typically takes more than a day. One of the desired state-of-the-art technologies would be the measurement of low average grades (a few ppm) of precious metals in real time and in situ during the various phases of mining exploration and production [3].

**Data availability statement:** All relevant data are within the manuscript and/or its Supporting Information files.

**Funding:** This work was primarily supported by the Natural Sciences and Engineering Research Council of Canada (NSERC) [grant number STPGP 521608-18]. Financial support was also provided by Impala Canada for the research.

**Competing interests:** The authors declare that they have no known competing financial interests or personal relationships that could appear to have influenced the work reported in this paper.

Laser-induced breakdown spectroscopy (LIBS) is a promising technology that can meet these requirements. LIBS is an optical analytical technique based on emission spectroscopy that uses a pulsed laser beam focused on the sample (solid, liquid or gas) to atomize a small area (typically less than $1\,mm^2$) and create a plasma. The light emitted from the plasma is then collected and spectrally analyzed. A schematic of the LIBS setup is shown in Fig 1. Reference materials are used to establish the relationship between the intensities of the spectral lines and the content of the analyte element of interest for each laser shot. The main potential advantages of LIBS over traditional analytical techniques lie in the ability to rapidly analyze samples with minimal or no preparation, regardless of the type of sample [4]. There are several reviews on the applications of LIBS in environmental and geochemistry [5–8]. LIBS has also been adopted or is being evaluated for its potential applications in several industries, including mineral processing, food, health, and archaeology [9].

In our recent work we have measured the concentration of palladium in cores from the Lac des Iles (LDI) palladium mine in Ontario, Canada by LIBS [10] and by LIBS assisted with laser fluorescence [11]. The cores used are of gabbronorite type and come from the same zone (B3) of the LDI mine. The cores were scanned with several thousand laser shots of approximately 750 µm diameter. Ore powders spiked with various concentrations of palladium chloride were used to calibrate our measurements. Our LIBS measurements indicate that the palladium distribution on the surface of the core samples is very inhomogeneous, varying by hundreds or even thousands of ppm between adjacent locations. We then estimated the mean core concentration of palladium by averaging the concentrations for all laser shots. For the cores analyzed, the LIBS measurements were found to be in good agreement with conventional analyses performed on the transverse half of the original cylindrical cores.

Although LIBS analysis of precious metal content in cores is in principle much faster than conventional laboratory methods, it remains important to optimize the duration of the LIBS analysis by performing appropriate sampling. The duration of LIBS measurements depends on the repetition rate of the pulsed laser, the number of sites analyzed on the sample surface, and the sampling strategy used. Therefore, a trade-off must be made between the duration of the measurement (i.e., the number

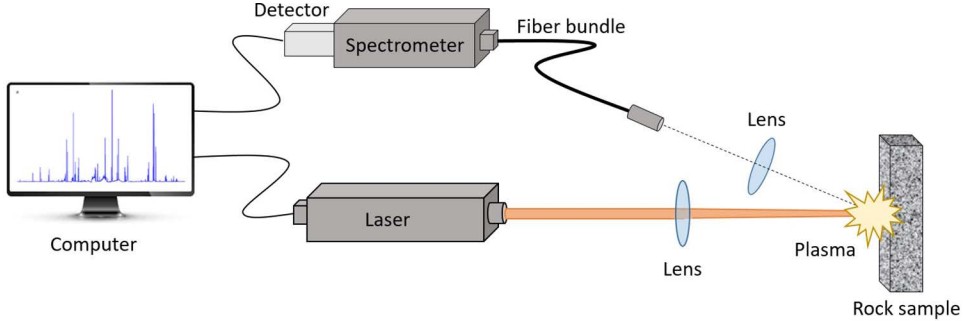

**Fig 1. Schematic of the LIBS experimental set-up.**

of laser shots) and the uncertainty in the mean concentration that is considered acceptable. An analogous issue has been discussed in [12–14] in the context of geochemical analysis of rocks containing mineral phases with concentrations in the wt% range and grain sizes comparable to the laser spot size. For such samples, it was found that as few as 10–15 [12,13] or 560 [14] laser shots at different positions were sufficient to determine their mineralogical composition.

In this paper we discuss the appropriate number of LIBS measurements to assess the palladium concentration at the ppm level in drill cores with a palladium grain size much smaller than the laser spot size. The approach we use is to take as a reference the distribution of palladium concentrations from drill cores of palladium ore from the LDI mine, scanned by LIBS at $M$ different positions, where $M$ is of the order of $10^4$. The measured palladium distributions in the analyzed cores are presented in Sect. 2. In Sect. 3.1, we consider subsets of $N$ measurements from this set of $M$ measurements and calculate the probabilities of obtaining the mean concentration $\mu_N$ of $N$ measurements within certain concentration limits around the reference concentration $\mu_M$ from the set of $M$ measurements. Then, in Sect. 3.2, we mathematically model the concentration distribution as well as the noise inherent in such measurements in a way that mimics the measured distribution for the set of $M$ measurements. This allows us to study a wider variety of palladium and noise distributions. Sect. 4 concludes the paper.

## 2 Measurements

The three cores from the LDI palladium mine considered in this study, hereafter referred to as core A, B and C, are quarter cores cut longitudinally, approximately 1 m long and 2 cm in radius. Their LDI designations are listed in S1 Table. The three cores are of gabbronorite type, and the major phases identified by μ-XRF, polarized light microscopy, and electron probe microanalysis are (1) calcium-rich plagioclase feldspar (mainly bytownite), (2) amphibole (mainly hornblende), and (3) sulfides (mainly chalcopyrite, pentlandite, pyrrhotite, and pyrite) [15]. The diameter of the laser spot on the target was approximately 750 μm, and the analyzed areas were separated by 1 mm in both directions. The size of platinum group minerals in the LDI mine is known to be less than a few tens of μm [16], which is much smaller than the laser spot size. Figs 2a and 2b show examples of LDI core fragments as delivered from the mine site.

Details of the laser parameters and experimental conditions used for the LIBS analysis of cores A and B are given in our previous work [10]. Briefly, the laser pulses generated by a Nd:YAG laser had a duration of 8 ns and a wavelength of 1064 nm. The laser fluence on the target was approximately 18 J cm$^{-2}$, the used acquisition delay and gate width were 4 and 10 μs, respectively, and the measurements were performed in ambient air. For these parameters, each laser shot was found to ablate $10^{-4}$–$10^{-3}$ mm$^3$ of material at the surface of the rock, depending on the mineral phase [17]. The palladium

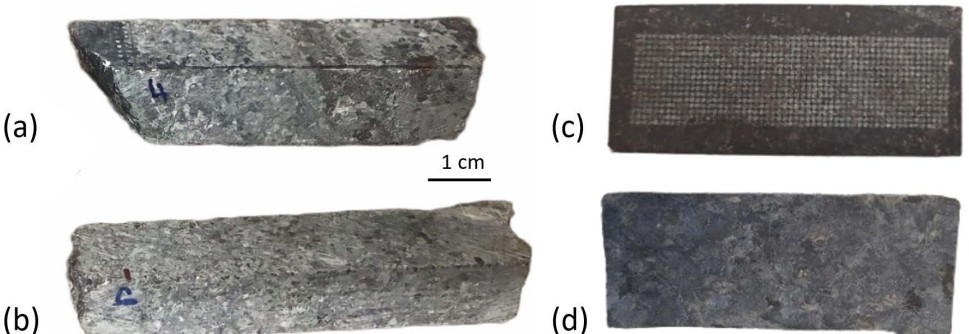

**Fig 2. Photos of quarter core fragments from the Lac des Iles palladium mine.** (a) and (b): Examples of core fragments as delivered from the mine site. (c): Example of a fragment face from core A after laser scanning on an 18×60 matrix (contrast enhanced). (d): Flat side of the flap, corresponding to (c), after trimming the rounded side of the fragment. The 1 cm scale applies to all 4 examples.

concentration for each laser shot was determined using 3 univariate calibration curves obtained from 3 sets of 6 reference materials each, as described in [10]. Fig 2c shows an example of a laser-scanned face of a fragment from core A, while Fig 2d shows the mirror image of this face, created by cutting the rounded side of the fragment to produce an additional flat surface. Core C was analyzed with slightly different laser parameters. The fluence was 14 J cm$^{-2}$, and the acquisition delay was 3 μs. In addition, only one calibration curve obtained from a set of 6 reference samples was used.

Fig 3 shows a raw (un-normalized) spectrum obtained from a laser shot at core A, centered on the Pd I 348.12 nm line used for palladium concentration determination. The spectrum is characterized by strong emission lines from iron and nickel, both high-emission elements. Background noise is also present due to inherent fluctuations in plasma emission, electronic variations, and the ICCD camera. In this case, the palladium line is clearly distinguishable from the background noise. However, when the palladium concentration is lower, the net palladium intensity $I_{Pd,net}$, calculated as the peak intensity minus the average background (represented by the dashed line), can approach the noise level. In some cases, $I_{Pd,net}$ may even become negative due to random noise fluctuations around the background level. This noise limitation ultimately defines the palladium detection limit of our LIBS system, which is estimated to be about 5 ppm [10]. Since the calibration curves exhibit a linear relationship $C_{Pd} \propto I_{Pd,net}$, negative $I_{Pd,net}$ values translate into negative $C_{Pd}$ values. Although negative concentrations are physically meaningless, they are included in the analysis for statistical accuracy. These negative values compensate for the excess positive concentrations introduced by noise, ensuring an accurate evaluation of average concentrations.

Fig 4 shows a portion of the concentration distribution of the three cores considered in this study as determined by LIBS. The results of the LIBS analyses are summarized in Table 1, which shows the number of laser shots performed on the three cores, as well as the mean concentration and standard deviation calculated from the distributions shown in Fig 4. Due to our laboratory setup, we were only able to scan the flat surfaces of the cores. Only the two flat surfaces of quarter cores B and C were scanned. For core A, we cut the round surface to form a third flat surface, which was also laser-scanned (see Fig 2c). The number of laser shots is not the same for the three cores due to this reason and the condition of the core fragments, which allowed more or less large rectangular laser scan matrices. While the concentration range in Fig 4 is limited to 1000 ppm for better visibility, the highest concentrations reach a

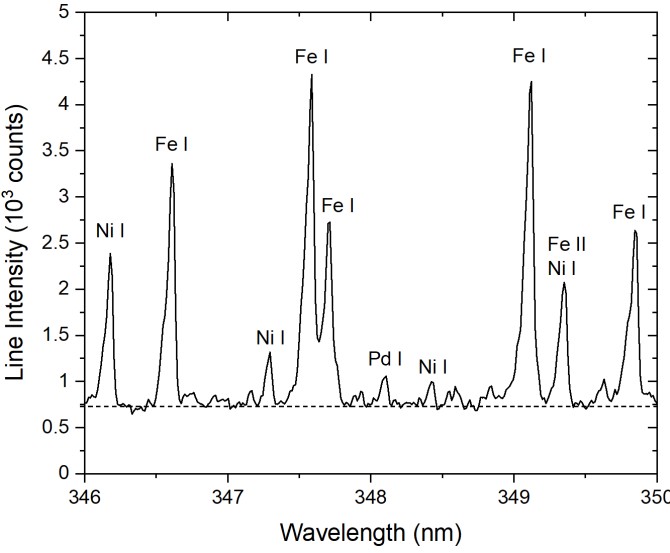

**Fig 3. Example of a raw spectrum around the Pd I 348.12 nm line.** The dashed line represents the average background emission.

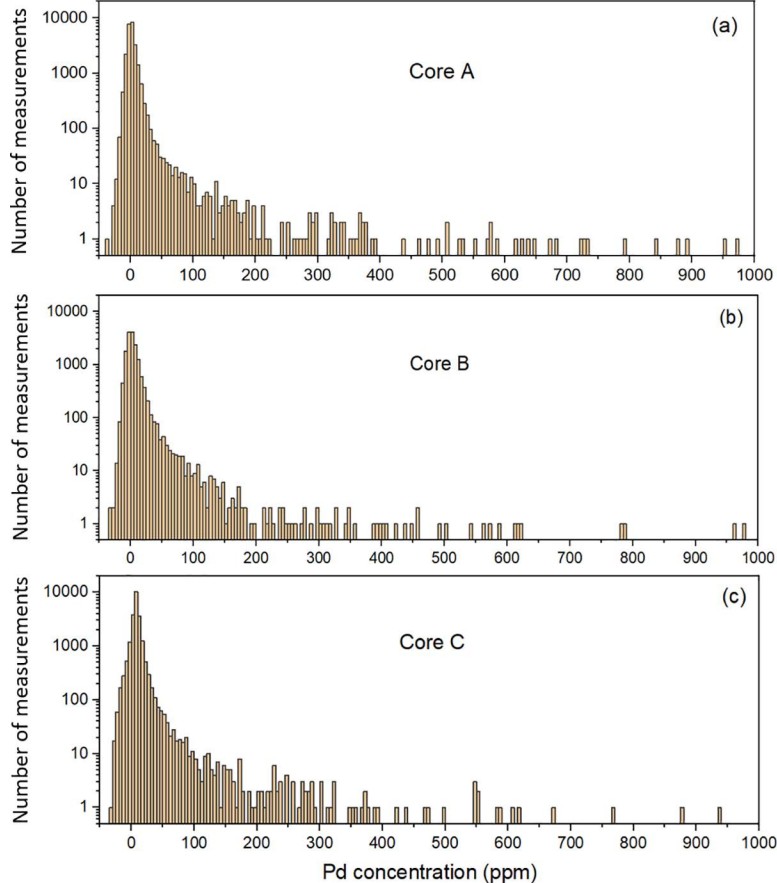

**Fig 4. Experimental distribution of palladium concentration.** Palladium distribution in ore from the LDI palladium mine for the three cores considered in this study. The bin size of the histograms is 5 ppm.

few thousand ppm for the three cores. The complete datasets of measurements for the three cores are provided in S2 Datasets. Note the presence of negative concentrations resulting from the extension of the calibration curve to negative values of $I_{Pd,net}$, as discussed above. The mean palladium concentration has been calculated taking into account these negative concentrations, which are offset by the part of the positive concentrations also due to noise, as discussed in detail in Sect. 3.2.

For all three cores, the LIBS measurements are in fairly good agreement with those determined by a certified laboratory using conventional methods for the longitudinal half of the original cylindrical cores, which are 4.9 ppm for core A, 7.7 ppm for core B and 12.8 ppm for core C. This agreement has been achieved despite the fact that LIBS performs a surface analysis, whereas wet chemical methods are applied to samples ground to grains of approximately 75 μm.

**Table 1. Results of LIBS analysis of the three quarter cores from the LDI mine considered in this study.**

| Core | Number of laser shots, $M$ | Mean Pd concentration, $\mu_M$ (ppm) | Standard deviation, $\sigma_M$ (ppm) |
|------|---------------------------|--------------------------------------|--------------------------------------|
| A | 25 165 | 5.4 | 49 |
| B | 16 030 | 6.5 | 42 |
| C | 22 698 | 10.1 | 29 |

## 3 Probability calculations

### 3.1 Probability calculations using experimental measurements

In this section, we generate many random sets of $N$ values, where $N$ ranges from 125 to 7 000, from among the $M$ experimental measurements made on the cores, and we use them to compute the probabilities of finding the mean concentration $\mu_N$ of any single set of $N$ measurements within predefined limits. To generate random sets of $N$ values among the $M$ experimental measurements, we used the xoshiro256** pseudorandom number generator with a period of $2^{256} - 1$, as implemented in the GNU Fortran compiler.

Two types of concentration bounds are considered. First, we consider the lower ($C_5$) and upper ($C_{95}$) concentrations corresponding to a 90% probability of obtaining a mean concentration between these two values, with 5% probabilities at either end of obtaining a mean concentration $\mu_N$ outside this interval. Second, we consider the probabilities of obtaining a mean concentration $\mu_N$ within an interval of ± 30% around the mean concentration $\mu_M$ of the $M$ measurements.

A key concept in this study is the Central Limit Theorem (CLT) of probability theory. Put simply, suppose the concentration distribution is characterized by a mean $\mu$ and a standard deviation $\sigma$. According to the CLT, the mean concentration distribution of a large number of randomly and independently selected sets of $N$ measurements will tend toward a normal (Gaussian) distribution with mean $\mu$ and standard deviation $\sigma N^{-1/2}$ as $N$ increases. The consequence of the CLT is that a larger number of measurements provides a greater accuracy in the mean concentration $\mu_N$ (i.e., smaller values of $\sigma N^{-1/2}$) and a larger standard deviation $\sigma$ requires more measurements to achieve a given accuracy. High values of $\sigma$ may be associated with the presence of local high concentrations (nuggets), in the case of a trace element that is predominantly present in discrete minor phase particles.

We generated the sets of $N$ elements in a completely random way, without the constraints that each element appears only once in a given set and that all sets are different. This is equivalent to extending the set of $M$ elements by an infinite replication of itself. In this way, $M^N$ sets of $N$ elements can be formed.

Fig 5 shows the normalized distribution obtained from the mean concentrations $\mu_N$ of $10^6$ randomly and independently generated sets of $N = 5\,000$ measurements on core A. The mean of this distribution is $\langle\mu_N\rangle = 5.4$ ppm, exactly the mean $\mu_M$ of the set of $M = 25\,165$ measurements, and its standard deviation is $0.70$ ppm, the same value as $\sigma_M N^{-1/2} = 0.70$ ppm expected from the CLT. As also expected from the CLT, the obtained distribution for $\mu_N$ is close to a normal distribution, in contrast to the distribution of the $M$ measurements (Fig 4a). However, the strong skewness of the latter does not lead to a true normal distribution for $N = 5\,000$. The concentrations $C_5$ and $C_{95}$ with 5% and 95% cumulative probability, respectively, are also shown. It follows that the mean concentration of any randomly selected $N$ measurements has a 90% probability of being between $C_5$ and $C_{95}$, with a 5% probability at either end of obtaining a mean concentration outside this range.

Fig 6 shows $C_5$ and $C_{95}$ for different numbers of measurements $N$, between 125 and 7 000, for $10^6$ random sets of $N$ measurements for cores A, B and C. The value of $10^6$ was chosen here to obtain reproducible results when repeating the calculations with a different seed in the random number generator. For all values of $N$ we find negligible differences between $\langle\mu_N\rangle$ and $\mu_M$ for the three cores. It can be seen that the gap between $C_5$ and $C_{95}$ narrows as $N$ increases due to the decrease in the standard deviation as $\sigma_M N^{-1/2}$. In addition, $C_5$ and $C_{95}$ become increasingly symmetric about $\mu_M$ as $N$ increases, i.e., as the distribution becomes closer to a normal distribution. For a normal distribution, $C_5$ and $C_{95}$ slowly converge to $\mu_M$ proportional to $N^{-1/2}$. Looking at core A, for $N = 7\,000$, $C_{95} - \mu_M = 1.03$ ppm and $\mu_M - C_5 = 0.90$ ppm which means that for any randomly distributed $N = 7\,000$ measurements, there is a 90% chance that the mean concentration $\mu_N$ has an error between +19% and −17% of the mean concentration $\mu_M = 5.4$ ppm.

An important practical parameter to consider is the lower concentration limit for ore exploitability which depends on the economics of the extraction process. For the different mineralized zones at the LDI mine, the palladium cut-off grade varies from 0.8 to 1.8 ppm [18]. In this work, we set this cut-off to $C_{min} = 1$ ppm. By definition, the parameter $C_5$ represents the threshold concentration such that there is only a 5% chance that the mean concentration $\mu_N$ for $N$ measurements will

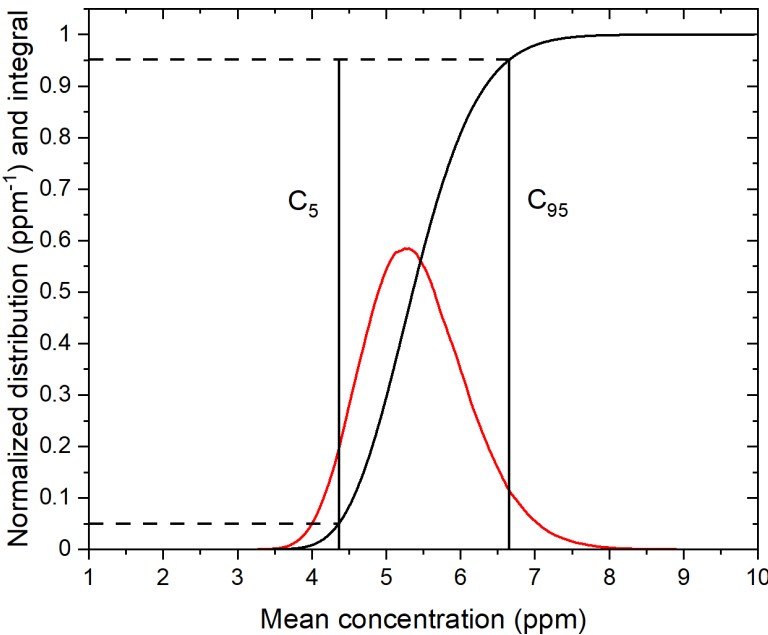

**Fig 5. Normalized mean concentration distribution.** Mean concentration distribution $\mu_N$ obtained from $10^6$ random sets of $N = 5\,000$ measurements (red curve) using the experimental data for core A. The black curve is the cumulative probability (integral of the red curve as a function of the mean concentration). The concentrations $C_5 = 4.36$ ppm and $C_{95} = 6.64$ ppm correspond to 5% and 95% of the cumulative probability, respectively.

fall below this value. Fig 6 shows that it would take less than $N = 125$ randomly distributed measurements (i.e., such that $C_5 > C_{min}$) to confirm with 95% confidence that the ore is suitable for processing.

In the event that an absolute measurement of the palladium concentration is required, we have also calculated the probability $P_{30}$ of obtaining a mean concentration $\mu_N$ within ± 30% of $\mu_M$. This error value seemed to us to be a reasonable choice for obtaining a useful estimate of the mean concentration. Fig 7 shows the probability $P_{30}$ as a function of the number of measurements $N$ for $10^6$ sets of $N$ measurements. We can see that $P_{30}$ quickly approaches 100% as the number of measurements $N$ increases. Assuming that the distribution is close to normal for large values of $N$, and using the properties of the error function, we can show that

$$100\% - P_{\gamma \times 100} \approx \frac{1}{\sqrt{\pi}\beta} \exp\left[-\beta^2\right],$$

(1)

when $\beta > 1$, where $\beta = \gamma \mu_M / \left(\sqrt{2} \sigma_M N^{-1/2}\right)$, the approximation improving as $\beta$ increases. Therefore, a smaller value of $\sigma_M / \mu_M$ favors a faster convergence of $P_{\gamma \times 100}$ to 100% as $N$ increases. This explains the comparative convergence rate of cores A, B, and C, since $\sigma_M / \mu_M = 9.1$, 6.7, and 2.9, respectively.

## 3.2 Mathematical model

In this section we discuss the approach of using an analytical concentration distribution instead of a particular set of experimental data as in the previous section. This allows one to establish a conceptual framework without being restricted to a particular set of $M$ measurements, to clarify the uncertainties associated with a finite set of $M$ measurements, to understand the effects of noise in the measurements, and to compute probabilities for arbitrary mean concentrations $\mu$. We will focus mainly on core A, since measurements were made on three faces of the core, and the number of measurements is higher than for cores B and C, and therefore likely to be more representative of the actual distribution of

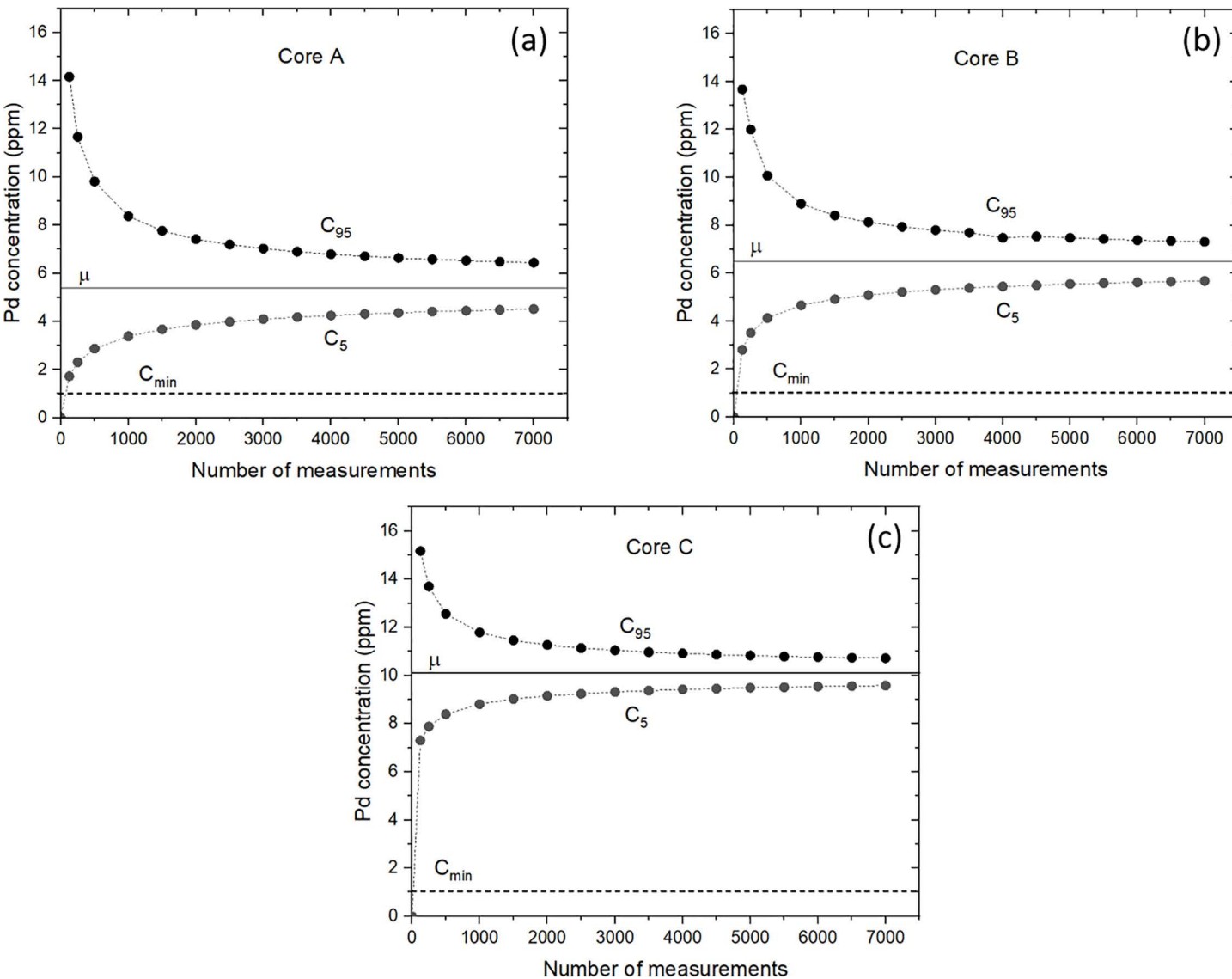

**Fig 6. Palladium concentrations vs. number of measurements for cores A, B and C.** $C_5$ and $C_{95}$ correspond to 5% and 95% of the cumulative probability, respectively, as a function of the number of randomly selected measurements $N$, for 106 sets of $N$ measurements. $\mu$ is the mean palladium concentration measured by LIBS (Table 1). The dashed line $C_{min} = 1\,\mathrm{ppm}$ represents the threshold concentration for the exploitability of the palladium ore.

palladium in the core. The basic Fortran code used to generate the theoretical palladium distributions is provided in S3 File.

Measurements, such as those shown in Fig 4, suggest that this analytical concentration distribution should have a steep slope at low concentrations and a gentle slope at high concentrations. The family of two-parameter functions of the form

$$f(x) = \exp\left[-\left(\frac{x}{\lambda}\right)^{\alpha}\right],$$

(2)

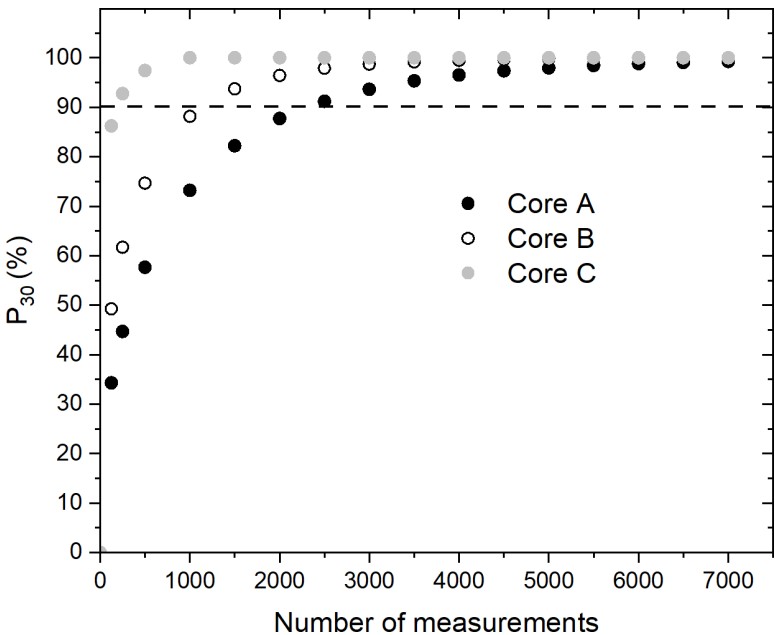

**Fig 7. Probability of obtaining the palladium concentration within ± 30% of the mean concentration $\mu_M$ vs. number of measurements.** Probability within ± 30% of the mean concentration $\mu_M$ as a function of the number of randomly selected measurements $N$, for $10^6$ sets of $N$ measurements. The dashed horizontal line represents the threshold for a probability ≥ 90%.

where $0 \leq x < \infty$ is the palladium concentration and $\lambda$ is a scaling parameter, meets these criteria provided $\alpha < 1$. The normalized function $\hat{f}(x) = f(x)/\int_0^\infty f(x)dx$ is the probability density (i.e., $\hat{f}(x)dx$ is the probability of measuring the concentration $x$ within an interval $dx$ containing $x$).

The mean concentration is given by

$$\mu = \int_0^\infty x\hat{f}(x)dx, \tag{3}$$

and the variance by

$$\sigma^2 = \int_0^\infty (x - \mu)^2\hat{f}(x)dx. \tag{4}$$

Here, the concentration variable $x$ is the continuous version of the discrete variable $x_i$, which is the result of the i-th laser shot. Each laser shot performs a local averaging over the area covered by the laser spot. In a loose analogy to the CLT, one might expect that this averaging process (which makes sense if the laser spot size is much larger than the grains containing palladium) would lead to a decrease in the variance of the empirical probability density $\hat{f}(x)$ as the laser spot size increases, but would not affect its mean value.

As discussed below, the value $\alpha = 1/10$ gives a palladium distribution of the studied core for $N = 25\,165$ virtual measurements similar to the experimental result, with the right balance between the low and high concentration populations. Smaller values of $\alpha$ increase the probability of obtaining higher concentrations, while larger values of $\alpha$ emphasize the low concentration population. However, no clear differences could be found between values of $\alpha$ around $1/10$. For $\alpha = 1/10$ we get the following exact results

$$\int_0^\infty f(x)\,dx = 3.6288 \times 10^6 \; \lambda,$$

(5)

$$\mu = 3.352212864 \times 10^{11} \; \lambda,$$

(6)

$$\sigma^2 = 2.425315246554341965824 \times 10^{25} \; \lambda^2.$$

(7)

Therefore, the ratio $\sigma/\mu$, which determines the rate of convergence of $P_{\gamma \times 100}$ to 100%, Eq. (1), is $14.69$ for $\alpha = 1/10$. For comparison, $\sigma/\mu = 8.67$ for $\alpha = 1/8$, $\sigma/\mu = 11.29$ or $\alpha = 1/9$, $\sigma/\mu = 19.10$ for $\alpha = 1/11$, and $\sigma/\mu = 24.82$ for $\alpha = 1/12$.

Fixing the mean palladium concentration at $\mu = 5$ ppm, approximately as in core A studied in the previous section, we find $\sigma = 73.45$ ppm. This value of $\sigma$ is higher than that obtained from the spectra measured with $M = 25\,165$ ($\sigma_M = 49$ ppm). However, as we will see below, the value of $\sigma_N$ can vary considerably from one set of $N = 25\,165$ virtual measurements to another. The probability density for these parameters is shown in Fig 8 (solid line).

To account for the noise inherent in the experimental measurements, we assumed a Gaussian concentration noise given by the probability density

$$\hat{f}_n(x) = \frac{1}{s_n \sqrt{2\pi}} \exp\left[ -\frac{1}{2} \left( \frac{x}{s_n} \right)^2 \right],$$

(8)

where $-\infty < x < \infty$ and $s_n$ is the standard deviation. Gaussian noise is generally considered a reasonable approximation for a wide variety of source noise in measurements. $s_n$ is expected to be comparable to the detection limit of palladium in our experiments, which we estimated to be 5–10 ppm per laser shot for core A [10].

The probability density for the palladium distribution including noise is given by the convolution

$$\hat{f}(x) = \int_0^\infty \hat{f}(x') \hat{f}_n(x - x')\,dx',$$

(9)

which is shown in Fig 8 for $s_n = 5$ ppm (dashed curve). It can be seen that the noise significantly smears $\hat{f}(x)$ around $x = 0$ but its effect becomes negligible for $x \; 25$ ppm.

Note that Gaussian noise does not affect the mean concentration $\mu$. If we define $\mu'$ as the mean concentration taking into account the noise, we have

$$\mu' = \int_{-\infty}^{\infty} x \hat{f}(x)\,dx = \int_0^\infty \hat{f}(x') \left( \int_{-\infty}^\infty x \hat{f}_n(x - x')\,dx \right)\,dx'.$$

(10)

The inner integral gives $x'$, from which we conclude that $\mu' = \mu$. This result justifies considering the entire experimental concentration distribution, including the negative part, when evaluating the mean palladium concentration. Similarly, we show that the variance of $\hat{f}(x)$ is the sum of the variances of the convolved functions (i.e., $\sigma'^2 = \sigma^2 + s_n^2$). In the presence of noise, the standard deviation of the mean concentrations of $N$ measurements becomes $\sigma' N^{-1/2}$. Note that these results hold not only for Gaussian noise, but for any function $\hat{f}_n(x)$ that is symmetric about $x = 0$ where $s_n^2$ is the variance of $\hat{f}_n(x)$.

To demonstrate the suitability of the probability density $\hat{f}(x)$ and its chosen parameters for modeling distributions such as those of Fig 4, we randomly performed $N$ virtual measurements in a manner similar to that done with the experimental data in Sect. 3.1. To do this, we first solved the following equation for the noiseless palladium concentration $x_{Pd}$

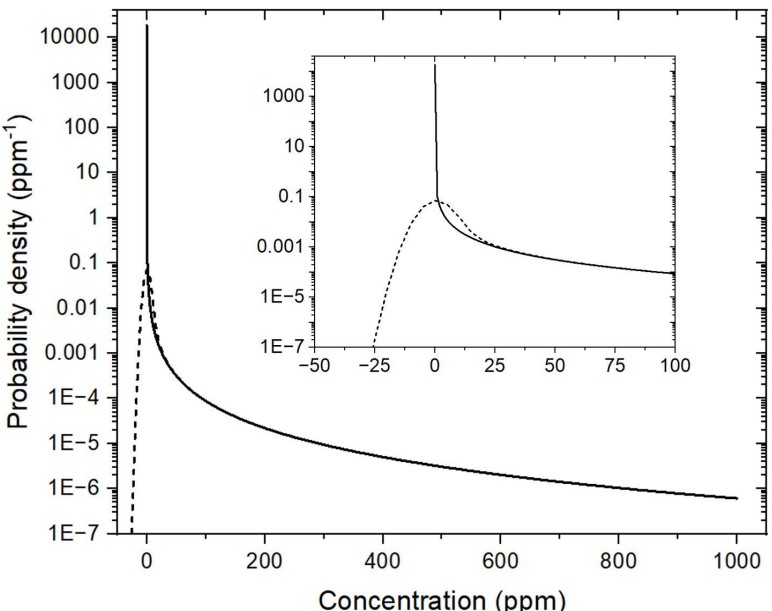

**Fig 8. Probability densities as a function of palladium concentration.** Solid curve: $\hat{f}(x)$ calculated from Eq. (1) with $\alpha = 1/10$ and $\lambda = 1.49 \times 10^{-11}$ ppm ($\mu = 5$ ppm). Dashed curve: convoluted function $\hat{f}'(x)$ calculated from Eq. (9) with $s_n = 5$ ppm. The inset is an enlargement around $x = 0$.

$$\int_0^{x_{Pd}} \hat{f}(x')dx' = y, \tag{11}$$

where $y$ is a random variable uniformly distributed in [0, 1[. For the function $\hat{f}(x)$ with $\alpha = 1/m$, this equation becomes

$$1 - e^{-z}\sum_{k=0}^{m-1}\frac{z^k}{k!} = y, \tag{12}$$

where $z = \left(\frac{x_{Pd}}{\lambda}\right)^{1/m}$.

The noise $x_n$ for each measurement of the palladium concentration is given by

$$\int_{-\infty}^{x_n} \hat{f}_n(x')dx' = y', \tag{13}$$

where $y'$ is a random variable uniformly distributed in [0, 1]. For Gaussian noise, Eq. (8), this equation becomes

$$\frac{1}{2}\left(1 + \mathrm{erf}\left[\frac{x_n}{s_n\sqrt{2}}\right]\right) = y'. \tag{14}$$

From a computational point of view, for a random number $y$, a concentration $x_{Pd} \geq 0$ is calculated from Eq. (11), and then for a random number $y'$ a noise-related concentration $x_n$ from Eq. (14), which can be positive or negative, is added to $x_{Pd}$, so that the concentration is $x = x_{Pd} + x_n$. In the following calculations we solved Eqs. (11) and (14) by the Newton-Raphson method for each set of $y$ and $y'$ with an accuracy of less than $2 \times 10^{-7}$ ppm. Another parameter used in the calculations is the range of $x_n$ which we set as $-20s_n \leq x_n < 20s_n$ to ensure the convergence of the solution of Eq. (14).

Typical examples of the palladium concentration distribution obtained using the procedure described above are shown in Fig 9. The number of virtual measurements used in Figs 9a, 9b and 9c is equal to the number of experimental measurements performed on cores A, B, and C, respectively, as described in Sect. 3.1. There is an obvious similarity with the experimental distributions of Fig 4. Of course, there is a very large number of possible realizations, since each is generated from $N$ random numbers $y$ and $y'$, and the limit is mostly determined by the accuracy of the solutions of Eqs. (11) and (14). In the case shown in Fig 9a, $\mu_N = 5.5$ ppm and $\sigma_N = 59$ ppm, in the case shown in Fig 9b, $\mu_N = 7.7$ ppm and $\sigma_N = 167$ ppm, while in the case shown in Fig 9c, $\mu_N = 10.5$ ppm and $\sigma_N = 135$ ppm. These values differ from $\mu$ and $\sigma'$ of the analytical probability densities ($\mu = 5$ ppm and $\sigma' = 74$ ppm for Fig 9a, $\mu = 7$ ppm and $\sigma' = 102$ ppm for Fig 9b, and $\mu = 10$ ppm and $\sigma' = 147$ ppm for Fig 9c) due to the limited sampling of $N$ virtual measurements.

We note some discrepancies between Fig 9 and Fig 4 around the zero concentration which is particularly noticeable for core C. This is the case for all realizations of $N$ virtual measurements tried. We could not find parameters within our three-parameter analytical model ($\alpha$, $\lambda$ and $s_n$) that gave a better fit to the experimental measurements. The model could possibly be improved by using a more general distribution for the noise, such as the generalized Student's distribution, which includes an additional parameter. A slightly higher value of $s_n$ was used in Fig 9b ($s_n = 7$ ppm) than in Figs 9a and

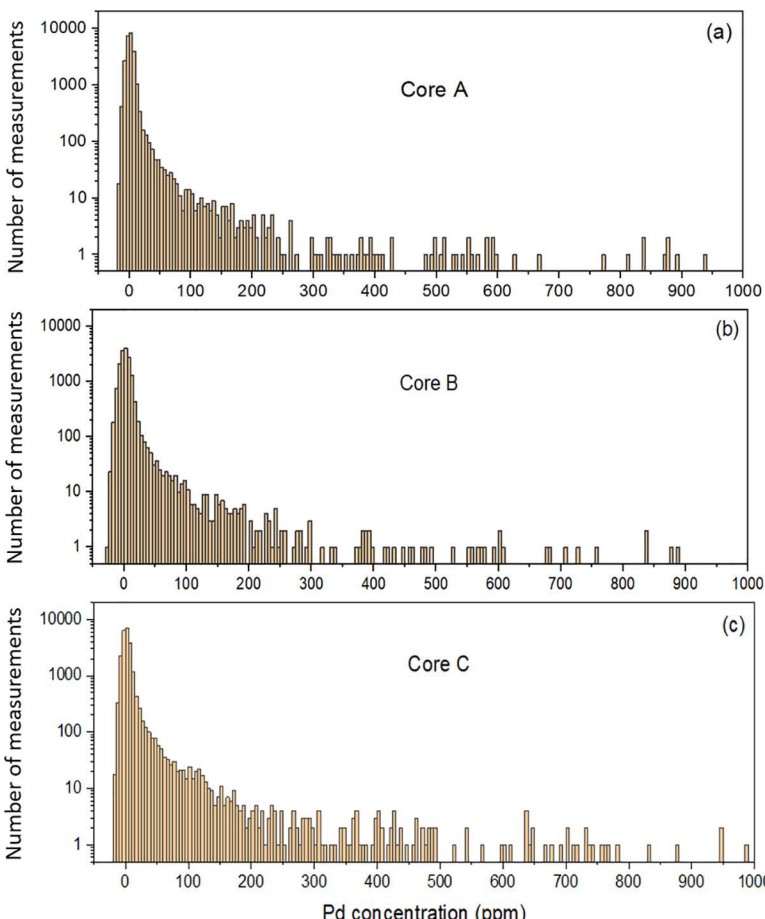

**Fig 9. Palladium concentration distributions using analytical probability density.** The model uses Eq. (2) with $\alpha = 1/10$ and $\lambda = \mu/3.352212864 \times 10^{11}$ and includes Gaussian noise given by Eq. (8). (a) $N = 25\,165$ virtual measurements with $\mu = 5$ ppm and $s_n = 5$ ppm. (b) $N = 16\,035$, $\mu = 7$ ppm and $s_n = 7$ ppm. (c) $N = 22\,698$, $\mu = 10$ ppm and $s_n = 5$ ppm. The bin size of the histograms is 5 ppm.

9c ($s_n = 5$ ppm) to improve the fit to Fig 4b. Although relatively small variations in $s_n$ have a noticeable effect on the concentration distribution near zero concentration, the effect of $s_n$ on the parameters of interest here, namely $C_5$, $C_{95}$ and $P_{30}$, is negligible as long as $(s_n/\sigma)^2 \ll 1$ since these parameters depend on $\sigma' = \left(\sigma^2 + s_n^2\right)^{1/2}$.

To have a closer look at the possible realizations of $N = 25\,165$ virtual measurements, we took the statistics of $10^6$ sets of $25\,165$ measurements. The mean of the values of $\mu_N$ obtained for each set of $N$ measurements is $\langle\mu_N\rangle = 5.00$ ppm with a standard deviation of $0.44$ ppm, the latter value being close to $\sigma' N^{-1/2} = 0.46$ ppm expected from the CLT. On the other hand, the mean of the values of $\sigma_N$ obtained for each set of $N$ measurements is $\langle\sigma_N\rangle = 62$ ppm with a standard deviation of $31$ ppm, and the most likely value is about $46$ ppm (close to the experimental value of $49$ppm). The distributions of $\mu_N$ and $\sigma_N$ are shown in Fig 10. As expected from the CLT, the obtained distribution for $\mu_N$ is close to a normal distribution in contrast to the probability density $\hat{f}(x)$. As suggested by additional calculations, the mean of the standard deviations $\langle\sigma_N\rangle$ would need a much higher value of $N$ to approach the expected value of $\sigma' = 74$ ppm. Fig 10 shows that different values of $\mu_N$ and $\sigma_N$ can be obtained from certain sets of $N = 25\,165$ measurements. Therefore, $N = 25\,165$ measurements may not always be representative of the intrinsic probability density $\hat{f}(x)$ of the sample. This may, of course, be the case for the experimental data shown in Fig 4.

### 3.3 Probability calculations using the mathematical model

Fig 11 shows $C_5$ and $C_{95}$ for different numbers of virtual measurements $N$ between 125 and 7 000 for $10^6$ sets of $N$ measurements. In the following calculations the bin size of the distributions has been set to 0.01 ppm. The result is similar to that of Fig 6 for the set of $M = 25\,165$ experimental measurements except that $C_{95} - C_5$ is somewhat larger here for the largest values of $N$ due to the larger value of $\sigma' = 74$ ppm compared to $\sigma_M = 49$ppm.

Our analytical model can be used to explore different scenarios. Still assuming $\alpha = 1/10$ (since there is no reason to make another assumption yet) and adjusting $\lambda$ to obtain the desired mean concentration $\mu$ using Eq. (6), Figs 12a and 12b show $C_5$ and $C_{95}$ for the cases $\mu = 1.5$ ppm and $\mu = 0.75$ ppm, respectively. In Fig 12a, we can see that $N = 2\,500$ measurements are required to confirm with 95% confidence that the analyzed sample is suitable for processing ($C_5 > C_{min}$).

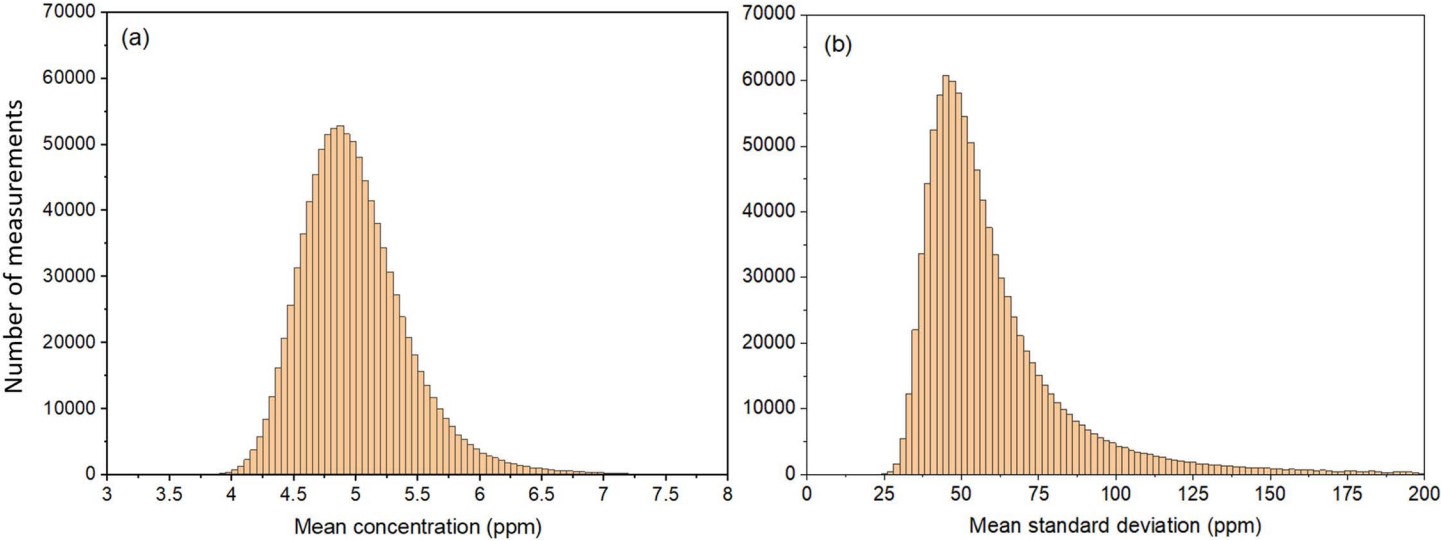

**Fig 10. Distributions of mean concentration and mean standard deviation.** Distributions of mean concentrations $\mu_N$ (a) and mean standard deviations $\sigma_N$ (b) for $10^6$ sets of 25 165 virtual measurements. The model uses Eq. (2) with $\alpha = 1/10$ and $\lambda = \mu/3.352212864 \times 10^{11}$ and includes Gaussian noise given by Eq. (8) with $s_n = 5$ ppm. The bin size of the histogram is 0.05 ppm in (a) and 2 ppm in (b).

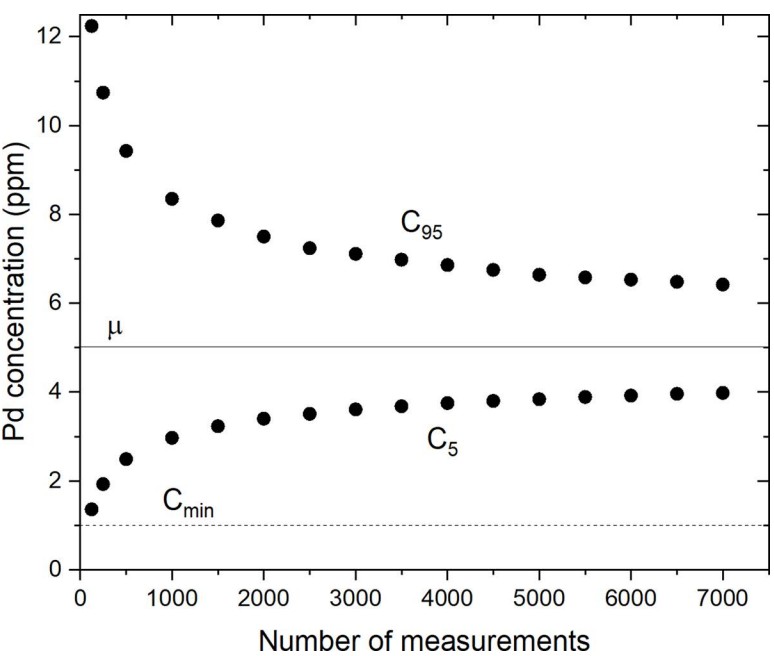

**Fig 11. Palladium concentrations vs. number of virtual measurements.** Palladium concentrations $C_5$ and $C_{95}$ at 5% and 95% cumulative probability, respectively, as a function of the number of randomly selected measurements $N$, for $10^6$ sets of $N$ measurements. The model uses Eq. (2) with $\alpha = 1/10$ and $\lambda = \mu/3.352212864 \times 10^{11}$ and includes Gaussian noise given by Eq. (8) with $s_n = 5$ ppm. The mean concentration $\langle \mu_N \rangle$ is practically $\mu = 5$ ppm for all values of $N$.

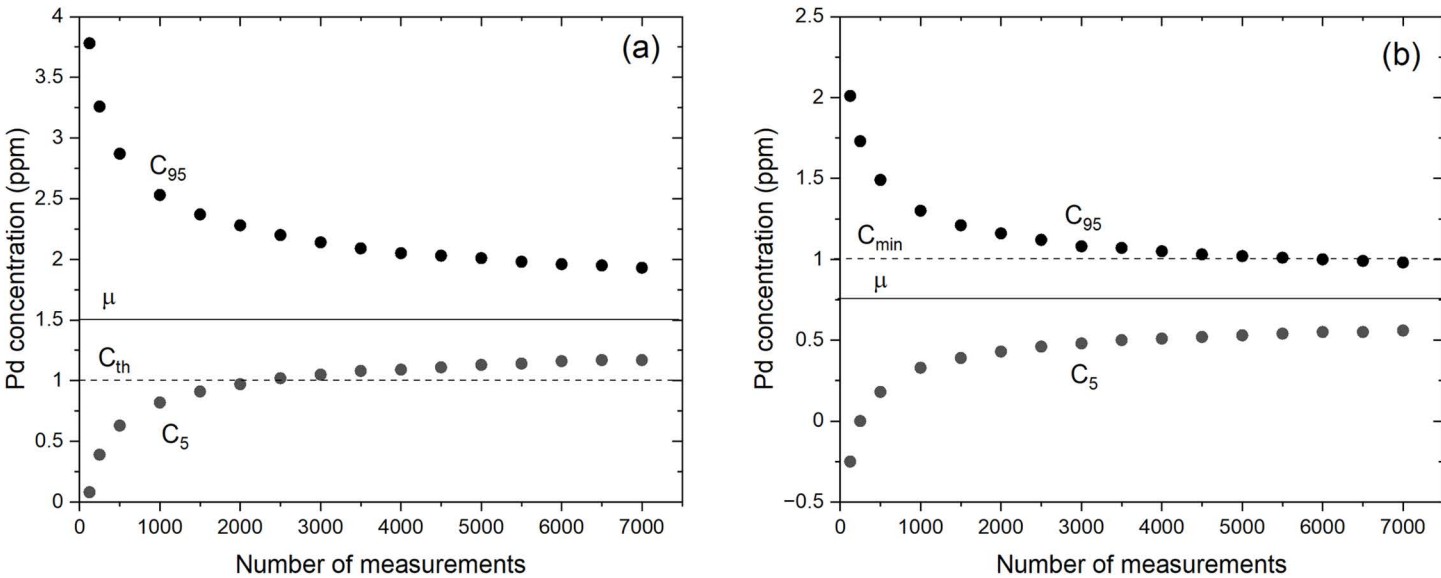

**Fig 12. Palladium concentrations vs. number of virtual measurements for different parameters $\lambda$.** Same as Fig 11 but for the parameter $\lambda$ adjusted so that $\mu = 1.5$ ppm (a) and $\mu = 0.75$ ppm (b).

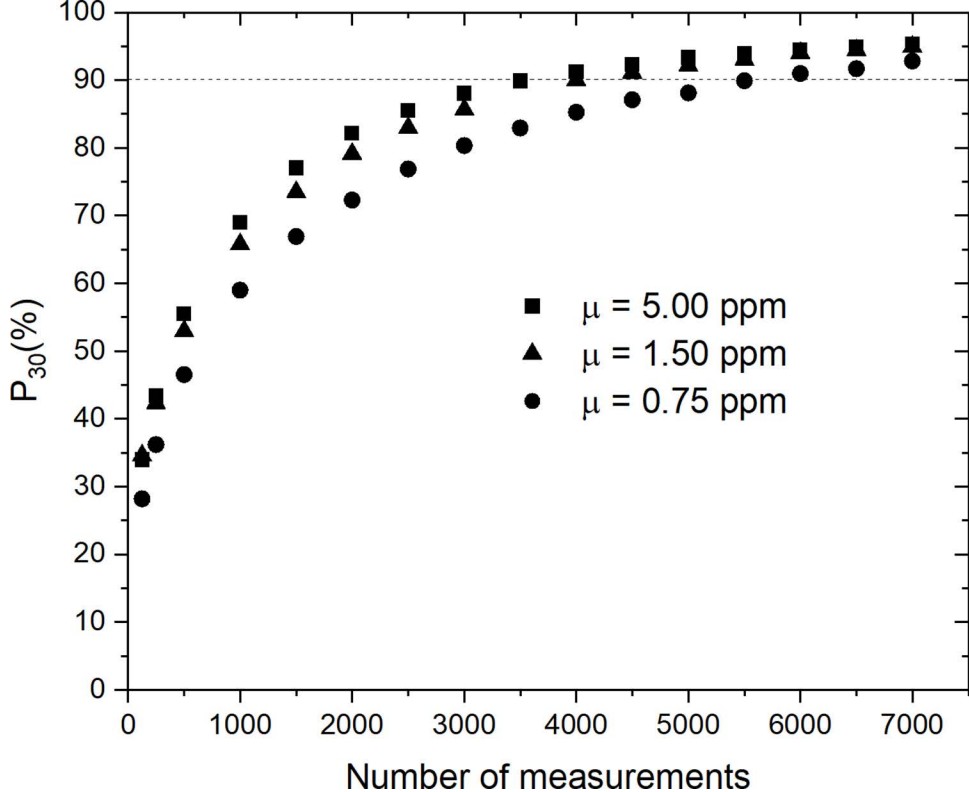

**Fig 13. Probability of obtaining a palladium concentration within ± 30% of the mean concentration $\mu$ vs. number of virtual measurements.**
Probability of obtaining a palladium concentration within ± 30% of the mean concentration $\mu$ as a function of the number of randomly selected measurements $N$, for $10^6$ sets of $N$ measurements. Same as Fig 11 but for the parameter $\lambda$ adjusted so that $\mu = 5.0, 1.5$ and $0.75$ ppm.

In Fig 12b $\mu < C_{min}$. In this case, more than $N = 5\,500$ virtual measurements are required to confirm with 95% confidence that the ore is not suitable for processing at all ($C_{95} < C_{min}$). In the case where $\mu \approx C_{min}$, no decision can be made based on these probabilistic considerations regardless of the number of measurements. However, any analytical method will run into difficulties in assessing the economic viability of the ore when the mean concentration is close to $C_{min}$. In the three cores examined at $\mu = 0.75,\ 1.50$ and $5.00$ ppm, it appears that a few thousand randomly distributed measurements are sufficient to make a decision.

Fig 13 shows the probability $P_{30}$ as a function of the number of measurements $N$ for $10^6$ sets of $N$ measurements for $\mu = 0.75,\ 1.50$ and $5.00$ ppm. The results are similar to those in Fig 7 except that the rate of convergence of $P_{30}$ to 100% is slower due to the higher value of $\sigma'/\mu = 14.8$.

## 4 Conclusion

In this paper a probabilistic study was carried out to estimate the number of LIBS measurements $N$ required to determine the mean palladium concentration within certain limits. Two types of limits were considered for each value of $N$. Firstly, the lower ($C_5$) and upper ($C_{95}$) concentrations corresponding to a 90% probability of obtaining a mean concentration between these two values, with a 5% probability at either end of obtaining a mean concentration outside this interval. In this case, if a given economic viability threshold ($C_{min}$) lies outside the interval $[C_5, C_{95}]$ then the sample can be considered suitable ($C_5 > C_{min}$) or unsuitable ($C_{95} < C_{min}$) for palladium extraction processing with 95% confidence. Second, an interval of ± 30% around the mean concentration $\mu$ in case an absolute concentration measurement is required. To perform this

probabilistic study, we first used a set of $M$ experimental measurements, assumed to be representative of the intrinsic palladium concentration probability density of the core, and then constructed an analytical probability density mimicking the $M$ measurements of the concentration distributions. The analytical function was used to explore the parameter space and to gain insight into the LIBS measurement process, in particular to understand the effect of noise on the measurements. The two approaches were found to give similar results despite the relatively small number of actual measurements made on the samples and the fact that only a relatively small fraction of the surface (less than 25%) was scanned.

The above analysis is a case study limited in scope to one trace element (palladium) in one type of ore (gabbronorite) from one specific area (B3) of the Lac des Iles palladium mine. The conclusion we draw from the analysis presented here is that a few thousand LIBS measurements, randomly distributed over the sample, are generally sufficient to assert that the average palladium concentration is within the confidence limits of practical interest. At a typical laser repetition rate of 50 Hz, 6 000 laser shots take 2 minutes, which is much faster than wet chemical analysis. As with any analytical method, more time would be required if greater precision were required. However, it should be remembered that LIBS performance depends on how representative the surface concentration is of the bulk core concentration.

Taking random laser shots at the core should not be a problem. In fact, the cylindrical core can be translated along its axis and rotated around its axis. By combining these two movements, the laser shots will form dotted spirals on the core. This allows the laser shots to be distributed relatively evenly across the surface of the core. In principle, with a fast processor, it would be possible to control the number of laser shots by monitoring the mean concentration trend during the analysis.

In order to go beyond the case study presented in this paper and provide useful guidelines for the use of LIBS in the mining industry, a systematic investigation of many representative samples is necessary to evaluate the appropriate LIBS methodology (laser parameters, type of reference materials, etc.) to be used for a given class of samples. An analytical probability density can be used to better understand the importance of certain parameters such as noise level, mean concentration and standard deviation as a function of instrumentation and ore composition. However, a large-scale application of the results of this study was beyond the scope of this paper and is left for future work.

## Supporting information

**S1 Table. Lac des Iles designation of the drill cores discussed in this work.**
(DOCX)

**S2 Datasets. Datasets of measurements for cores A, B and C.**
(ZIP)

**S3 File. Basic Fortran code used to generate the theoretical palladium distributions.**
(TXT)

## Acknowledgements

We are grateful to Lionnel Djon formerly from Impala Canada for providing us with quarter drill cores from the Lac des Iles mine and their laboratory analysis.

## Author contributions

**Conceptualization:** François Vidal.

**Data curation:** François Vidal, Samira Selmani, Ismail Elhamdaoui, Nessrine Mohamed.

**Formal analysis:** François Vidal.

**Funding acquisition:** François Vidal, Marc Constantin.

**Investigation:** François Vidal, Samira Selmani, Ismail Elhamdaoui.

**Methodology:** François Vidal.

**Project administration:** François Vidal, Marc Constantin, Mohamad Sabsabi.

**Resources:** François Vidal, Paul Bouchard, Marc Constantin, Mohamad Sabsabi.

**Software:** François Vidal.

**Supervision:** François Vidal, Marc Constantin, Mohamad Sabsabi.

**Validation:** François Vidal.

**Visualization:** François Vidal.

**Writing – original draft:** François Vidal.

**Writing – review & editing:** François Vidal, Samira Selmani, Ismail Elhamdaoui, Marc Constantin, Mohamad Sabsabi.

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
