## [Decision Letter · Decision Letter 0]

11 Dec 2024

PONE-D-24-50584Assessment of palladium concentration in drill cores using laser-induced breakdown spectroscopy (LIBS)PLOS ONE

Dear Dr. Vidal,

Thank you for submitting your manuscript to PLOS ONE. After careful consideration, we feel that it has merit but does not fully meet PLOS ONE’s publication criteria as it currently stands. Therefore, we invite you to submit a revised version of the manuscript that addresses the points raised during the review process.

We look forward to receiving your revised manuscript.

Kind regards,

Amit Kumar Goyal, PhD

Academic Editor

PLOS ONE

Journal Requirements:

2. Thank you for stating the following financial disclosure: [This work was primarily supported by the National Science and Engineering Research Council of Canada (NSERC) [grant number STPGP 521608-18]. Financial support was also provided by Impala Canada].

Reviewers' comments:

Reviewer's Responses to Questions

**Comments to the Author**

1. Is the manuscript technically sound, and do the data support the conclusions?

Reviewer #1: Yes

Reviewer #2: Yes

2. Has the statistical analysis been performed appropriately and rigorously? 

Reviewer #1: Yes

Reviewer #2: No

3. Have the authors made all data underlying the findings in their manuscript fully available?

Reviewer #1: Yes

Reviewer #2: Yes

4. Is the manuscript presented in an intelligible fashion and written in standard English?

Reviewer #1: Yes

Reviewer #2: Yes

5. Review Comments to the Author

Reviewer #1: It is recommended to include LIBS spectra in this article, improving discussion based on the empirical evidence in which obtained through the LIBS method. The emprical data could be useful to support whereas the LIBS method is more effective than conventional methods regarding Pd analysis.

Reviewer #2: The manuscript reports measurements of palladium concentration in drill cores using Laser-Induced Breakdown Spectroscopy (LIBS). The subject matter is interesting and relevant, particularly in addressing the number of minimum measurements required for a reliable assessment of palladium concentration. However, the presentation lacks clarity, and several critical aspects of the manuscript require significant improvement before it can be considered for publication.

1. The rationale behind the choice of varying numbers of measurements for three different cores is not explained. It is crucial to justify these experimental parameters.

2. The inclusion of measurements showing a negative error in concentration estimation is highly questionable. In effect the cumulative negative errors result in nullifying a certain number of positive measurements. The details on why negative values are obtained and the rational for their inclusion is not clear. Also, it will be worth mentioning the percentage of these measurements. How will the results change if only the positive values are considered.

3. The theoretical section is not clearly connected to the experimental results. The authors must explicitly state the purpose of this section, its relevance to the current study, and its potential implications for future experiments. How does this study enhance the understating, particularly for future measurements/Does it help in reduced no of measuserments? Also this section is too lengthy.

4. Figure 4 is not readable and does not effectively convey the results. It would be more appropriate to present the data in three separate graphs for clarity. Additionally, the mean concentration should be clearly indicated in each graph to enhance interpretability.

6. PLOS authors have the option to publish the peer review history of their article (what does this mean? ). If published, this will include your full peer review and any attached files.

**Do you want your identity to be public for this peer review?** For information about this choice, including consent withdrawal, please see our Privacy Policy .

Reviewer #1: No

Reviewer #2: No

---

## [Author Response · Author response to Decision Letter 0]

23 Jan 2025

Reviewer #1: It is recommended to include LIBS spectra in this article, improving discussion based on the empirical evidence in which obtained through the LIBS method. The empirical data could be useful to support whereas the LIBS method is more effective than conventional methods regarding Pd analysis.

We have incorporated a new Figure 2, which provides an example of a raw spectrum obtained at a specific position in core A. This figure also highlights the issue of noise in LIBS measurements, as clearly demonstrated in the displayed spectrum.

Regarding the efficiency of LIBS compared to conventional methods: as noted in the introduction, conventional techniques involve "a laborious, time-consuming, and energy-intensive sequence of grinding, homogenization, and dissolution, typically requiring more than a day." This estimate excludes the additional time needed for sample transport to often distant laboratories. In contrast, LIBS can rapidly scan the surface of a core on-site within minutes, with spectra processed in real time. However, as emphasized in the conclusion, LIBS is not intended to replace conventional methods where high accuracy is paramount. This limitation arises in part because LIBS is a surface analysis technique, whereas conventional methods analyze pulverized core samples, providing a more representative average composition.

Reviewer #2: The manuscript reports measurements of palladium concentration in drill cores using Laser-Induced Breakdown Spectroscopy (LIBS). The subject matter is interesting and relevant, particularly in addressing the number of minimum measurements required for a reliable assessment of palladium concentration. However, the presentation lacks clarity, and several critical aspects of the manuscript require significant improvement before it can be considered for publication.

1. The rationale behind the choice of varying numbers of measurements for three different cores is not explained. It is crucial to justify these experimental parameters.

The number of measurements taken for the three cores varied depending on their structural quality and specific circumstances. For example, the rounded section of core A was cut to create a third flat surface, ensuring a more representative sampling of the core. This modification was performed as part of the detailed study reported in Reference 10 (Selmani et al., 2022), but it was not repeated for cores B and C due to the time and complexity involved. Additionally, all cores were delivered in multiple fragments, some of which were broken at the sides. As a result, only smaller, rectangular laser shot arrays could be applied to certain fragments within our laboratory setup. This point is addressed in the revised manuscript.

2. The inclusion of measurements showing a negative error in concentration estimation is highly questionable. In effect the cumulative negative errors result in nullifying a certain number of positive measurements. The details on why negative values are obtained and the rational for their inclusion is not clear. Also, it will be worth mentioning the percentage of these measurements. How will the results change if only the positive values are considered?

We agree with the reviewer that the concept of negative concentrations is meaningless when taken out of context. Their inclusion is merely a mathematical twist that simplifies the statistical treatment in our data. As the reviewer noted, negative concentrations counterbalance the excess of positive concentrations caused by noise, thus providing a more balanced and accurate analysis.

A discussion of the new Figure 2 of the revised manuscript, which shows a spectrum with spectral lines and noise, aims to address this point in more detail.

As discussed in section 3.4 of Reference 10, 88% of the data collected for core A showed palladium concentrations below 10 ppm, which means that slightly less than half of them (41%) are negative (most of them being very close to 0), given the approximate symmetry of the distribution around the zero concentration. Despite this 88%, the total contribution of concentrations below 10 ppm (including negatives) accounts for only 2% of the core’s average concentration, with the remaining 98% derived from local concentrations exceeding 10 ppm. This reflects the nuggety nature of palladium distribution in the ore. Excluding negative concentrations would skew the results by only capturing the excess positive concentrations introduced by noise. Conversely, using a threshold of >10 ppm to filter out noise partially addresses this issue but is not universally applicable, as the threshold may vary across datasets. Adding all concentrations, including the negatives, provides a safer and more accurate method for calculating the average concentration. This approach ensures that noise effects are effectively neutralized without introducing arbitrary thresholds.

3. The theoretical section is not clearly connected to the experimental results. The authors must explicitly state the purpose of this section, its relevance to the current study, and its potential implications for future experiments. How does this study enhance the understating, particularly for future measurements/Does it help in reduced no of measurements? Also this section is too lengthy.

In our view, the theoretical sections (Sections 3.2 and 3.3) are the most compelling parts of the paper, as they provide a comprehensive discussion that extends beyond the limitations of the available experimental data. We believe their length is appropriate given their significance, and their motivation is clearly articulated at the outset of Section 3.2:

"In this section we discuss the approach of using an analytical concentration distribution instead of a particular set of experimental data as in the previous section. This allows one to establish a conceptual framework without being restricted to a particular set of M measurements, to clarify the uncertainties associated with a finite set of M measurements, to understand the effects of noise in the measurements, and to compute probabilities for arbitrary mean concentrations μ."

This introduction provides the necessary context for the depth of analysis presented in these sections and answers the reviewer’s first question. Regarding the reviewer’s second question, the theoretical sections do not directly aim to reduce the number of measurements. Instead, they focus on optimizing the use of measurements by fostering a deeper understanding of potential sources of uncertainty. This allows for more informed interpretation and application of the experimental data, enhancing the reliability and robustness of the conclusions drawn. It also provides a model of the palladium distribution in the Lac des Iles ore.

4. Figure 4 is not readable and does not effectively convey the results. It would be more appropriate to present the data in three separate graphs for clarity. Additionally, the mean concentration should be clearly indicated in each graph to enhance interpretability.

Thank you for your advice. We have split the figure in question into three separate ones.

---

## [Decision Letter · Decision Letter 1]

11 Feb 2025

PONE-D-24-50584R1Assessment of palladium concentration in drill cores using laser-induced breakdown spectroscopy (LIBS)PLOS ONE

Dear Dr. Vidal,

Thank you for submitting your manuscript to PLOS ONE. After careful consideration, we feel that it has merit but does not fully meet PLOS ONE’s publication criteria as it currently stands. Therefore, we invite you to submit a revised version of the manuscript that addresses the points raised during the review process.

We look forward to receiving your revised manuscript.

Kind regards,

Amit Kumar Goyal, PhD

Academic Editor

PLOS ONE

Journal Requirements:

Reviewers' comments:

Reviewer's Responses to Questions

**Comments to the Author**

1. If the authors have adequately addressed your comments raised in a previous round of review and you feel that this manuscript is now acceptable for publication, you may indicate that here to bypass the “Comments to the Author” section, enter your conflict of interest statement in the “Confidential to Editor” section, and submit your "Accept" recommendation.

Reviewer #1: All comments have been addressed

Reviewer #3: All comments have been addressed

2. Is the manuscript technically sound, and do the data support the conclusions?

Reviewer #1: Yes

Reviewer #3: Yes

3. Has the statistical analysis been performed appropriately and rigorously? 

Reviewer #1: Yes

Reviewer #3: I Don't Know

4. Have the authors made all data underlying the findings in their manuscript fully available?

Reviewer #1: Yes

Reviewer #3: Yes

5. Is the manuscript presented in an intelligible fashion and written in standard English?

Reviewer #1: Yes

Reviewer #3: Yes

6. Review Comments to the Author

Reviewer #1: It will be better if the english style of the manuscript is improved, so that all scientific information and discussion based empherical data will be deliverd effectively

Reviewer #3: Thanks for addressing my comments. I think the manuscript can be published after a few minor tweaks. At this point, just one revision is needed: please include the pictures of drilling samples. If you can, it would be great to see images of the sample surface before and after LIBS detection.

7. PLOS authors have the option to publish the peer review history of their article (what does this mean? ). If published, this will include your full peer review and any attached files.

**Do you want your identity to be public for this peer review?** For information about this choice, including consent withdrawal, please see our Privacy Policy .

Reviewer #1: No

Reviewer #3: No

---

## [Author Response · Author response to Decision Letter 1]

19 Feb 2025

We thank the reviewers for taking the time to evaluate our manuscript and for sharing their concerns and recommendations.

At this point, only minor changes to the manuscript were recommended:

Reviewer #1: It will be better if the english style of the manuscript is improved, so that all scientific information and discussion based empherical data will be deliverd effectively

We are sorry that some parts of the paper seemed unclear to Reviewer #1. Although none of the authors are native English speakers, this is not a criticism we have typically received from reviewers of our other papers. Nevertheless, we have revised our manuscript using various software, including DeepL Write set to American English, and they recommended a few changes which will likely improve the English quality of the paper.

Reviewer #3: “Thanks for addressing my comments. I think the manuscript can be published after a few minor tweaks. At this point, just one revision is needed: please include the pictures of drilling samples. If you can, it would be great to see images of the sample surface before and after LIBS detection.”

We appreciate the idea of showing some photos of our work specimens, although more are shown in our previous publications. In the new Fig 2, we show photos of raw core fragments similar to those used in this work, and the laser-scanned surface of a fragment together with its mirror image obtained by trimming the round side of the fragment. Unfortunately, we did not have photos of the same surface before and after laser scanning.

We hope that this new version of our manuscript is to your satisfaction.

---

## [Editor Report · Decision Letter 2]

21 Feb 2025

Assessment of palladium concentration in drill cores using laser-induced breakdown spectroscopy (LIBS)

PONE-D-24-50584R2

Dear Dr. Vidal,

We’re pleased to inform you that your manuscript has been judged scientifically suitable for publication and will be formally accepted for publication once it meets all outstanding technical requirements.

Kind regards,

Amit Kumar Goyal, PhD

Academic Editor

PLOS ONE
---

## [Editor Report · Acceptance letter]

PONE-D-24-50584R2

PLOS ONE

Dear Dr. Vidal,

I'm pleased to inform you that your manuscript has been deemed suitable for publication in PLOS ONE. Congratulations! Your manuscript is now being handed over to our production team.

Kind regards,

on behalf of

Dr. Amit Kumar Goyal

Academic Editor

PLOS ONE